# Energy Efficient Transmission Design for NOMA Backscatter-Aided UAV Networks with Imperfect CSI

**Saad AlJubayrin [1], Fahd N. Al-Wesabi [2], Hadeel Alsolai [3], Mesfer Al Duhayyim [4], Mohamed K. Nour [5], Wali Ullah Khan [6], Asad Mahmood [6], Khaled Rabie [7,8,*] and Thokozani Shongwe [8,*]**

1   Department of Computer Science, College of Computing and Information Technology, Shaqra University, Shaqraa 11911, Saudi Arabia; aljubayrin@su.edu.sa
2   Department of Computer Science, College of Science & Art at Mahayil, King Khalid University, Abha 62529, Saudi Arabia; falwesabi@kku.edu.sa
3   Department of Information Systems, College of Computer and Information Sciences, Princess Nourah bint Abdulrahman University, Riyadh 11671, Saudi Arabia; hdalsoli@pnu.edu.sa
4   Department of Computer Science, College of Sciences and Humanities-Aflaj, Prince Sattam bin Abdulaziz University, Al-Kharj 16278, Saudi Arabia; msduhayyim@psau.edu.sa
5   Department of Computer Sciences, College of Computing and Information System, Umm Al-Qura University, Mecca 24382, Saudi Arabia; mnour@uqu.edu.sa
6   Interdisciplinary Centre for Security, Reliability and Trust (SnT), University of Luxembourg, 1855 Luxembourg, Luxembourg; waliullah.khan@uni.lu (W.U.K.); asad.mahmood@uni.lu (A.M.)
7   Department of Engineering, Manchester Metropolitan University, Manchester M15 6BH, UK
8   Department of Electrical and Electronic Engineering Technology, University of Johannesburg, Doornfontein, P.O. Box 17011, Johannesburg 2028, South Africa
*   Correspondence: k.rabie@mmu.ac.uk (K.R.); tshongwe@uj.ac.za (T.S.)

**Abstract:** The recent combination of ambient backscatter communication (ABC) with non-orthogonal multiple access (NOMA) has shown great potential for connecting large-scale Internet of Things (IoT) in future unmanned aerial vehicle (UAV) networks. The basic idea of ABC is to provide battery-free transmission by harvesting the energy of existing RF signals of WiFi, TV towers, and cellular base stations/UAV. ABC uses smart sensor tags to modulate and reflect data among wireless devices. On the other side, NOMA makes possible the communication of more than one IoT on the same frequency. In this work, we provide an energy efficient transmission design ABC-aided UAV network using NOMA. This work aims to optimize the power consumption of a UAV system while ensuring the minimum data rate of IoT. Specifically, the transmit power of UAVs and the reflection coefficient of the ABC system are simultaneously optimized under the assumption of imperfect channel state information (CSI). Due to co-channel interference among UAVs, imperfect CSI, and NOMA interference, the joint optimization problem is formulated as non-convex, which involves high complexity and makes it hard to obtain the optimal solution. Thus, it is first transformed and then solved by a sub-gradient method with low complexity. In addition, a conventional NOMA UAV framework is also studied for comparison without involving ABC. Numerical results demonstrate the benefits of using ABC in a NOMA UAV network compared to the conventional UAV framework.

**Keywords:** ambient backscatter communication (ABC); non-orthogonal multiple access (NOMA); Internet of Things (IoT); unmanned aerial vehicles (UAVs)

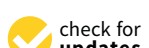



## 1. Introduction

The future sixth-generation (6G) systems are expected to connect massive devices, support low energy consumption, provide diverse quality of services, and experience very short latency [1]. These devices also contain low-powered wireless devices known as the Internet of Things (IoT) which provide small data rates. In this regard, some disruptive technology directions are intelligent reconfigurable surfaces [2], ambient backscatter communication [3], Tera-hertz communication [4], non-orthogonal multiple access (NOMA) [5], unmanned

aerial vehicles (UAVs) [6], and artificial intelligent/machine learning [7]. Among these two promising technologies are backscatter communication and NOMA, which provide very high energy and spectral efficiency. Using the existing radio frequency (RF) signals, i.e., WiFi, TV, cellular base stations, etc., backscatter communication is the concept of providing battery-free transmission among different wireless devices. Backscatter communication with the help of sensors harvests energy using these signals for circuit operation, then reflects the signal to IoST by modulating valuable data [8]. In addition, NOMA allows multiple sensors to simultaneously communicate over the same frequency. It can be achieved through superposition coding and successive interference cancellation (SIC) techniques. The integration of ABC with NOMA can support the challenging requirements of future 6G networks.

UAVs, also known as drones, perform autonomous flight missions without a human pilot onboard [9]. UAVs can be used for various applications such as rescue operations, aerial photography, cargo transport, firefighting, agriculture, and telecommunications. They are also used for military operations, i.e., surveillance, monitoring, and attacking hostile targets. UAVs have recently drawn significant attention due to their low acquisition and maintenance costs, hovering capability, low manufacturing cost, and high mobility [10]. A UAV can be used as a base station to provide cellular connections in flooded regions, hot-spot zones, and remote areas where cellular infrastructures are unavailable. It is also used as a relay node to improve the link connection between transmitter and receiver [11]. Recently, different works on NOMA and UAV communications have been reported. The authors of [12] have integrated NOMA with intelligent reflecting surfaces and optimized beamforming, power allocation, and phase design to improve the energy efficiency of the system. The work in [13] has minimized the power consumption by optimizing the UAV trajectory selecting an efficient cluster head for UAV-based sensor networks.

### 1.1. Recent Works on Backscatter Communications

Various research studies have been carried out on the performance of backscatter communication in single-cell terrestrial networks. For example, the work in [14] has provided a joint resource allocation approach to maximize the achievable energy efficiency of backscatter-aided NOMA networks. Jameel et al. [15] have employed a reinforcement learning algorithm for reliable and scalable backscatter communication in IoT networks. The work in [16] has investigated the system spectral efficiency by optimizing the power allocation of the source and reflection power of the backscatter sensor in backscatter-aided NOMA networks. To enhance the minimum throughput and ensure user fairness, the study in [17] has optimized the subcarrier assignment, time allocation, and reflection power in full duplex backscatter-aided NOMA systems. Another work in [18] has also considered full duplex backscatter communication IoT network to maximize the minimum throughput by optimizing the time allocation, subcarrier assignment, and reflection power of the system. Moreover, the authors of [19] have proposed a secure communication design for a backscatter-aided NOMA network in the presence of multiple eavesdroppers. Zhuang et al. [20] have investigated a new resource allocation problem in a cognitive radio-based backscatter-aided NOMA network to enhance the achievable throughput of the system. Xu et al. [21] have minimized the total energy consumption of a mobile edge computing-based backscatter communication network by optimizing user association, time allocation, transmit power, computational offloading, and reflection power. Furthermore, the work of [22] has studied a resource management problem to enhance the spectral efficiency of backscatter-aided NOMA networks. The work in [23] has investigated physical layer security in a cognitive ABC network. Of late, some researchers have also studied backscatter communication in multi-cell terrestrial networks. For instance, Ahmed et al. [24] have presented an efficient resource allocation scheme in backscatter-aided multi-cell NOMA networks to maximize the system achievable energy efficiency with imperfect SIC detection. The authors of [25] have exploited a learning-based algorithm for efficient resource allocation in backscatter-aided vehicular networks. Jameel et al. [26] have adopted reinforcement

learning for interference management in software-defined backscatter-aided heterogeneous networks.

Recently, some works have also considered backscatter in UAV networks. For instance, Hua et al. have maximized the throughput of backscatter-aided UAV networks through efficient time allocation, reflection coefficient, and trajectory design [27]. Farajzadeh et al. have explored a problem of UAV altitude and trajectory optimization to improve the successful decoded bits of NOMA backscatter-aided UAV networks [28]. The authors of [29] have maximized the throughput of backscatter-aided UAV networks through dynamic power splitting, reflection coefficient and trajectory design. Han et al. have instigated probabilities of error detection and bit error rate in backscatter-aided UAV networks [30]. They have also optimized the trajectory of UAVs for efficient power consumption. To improve the fair secrecy rate, Hu et al. have proposed a block coordinate descent approach to optimize the backscatter scheduling, UAV trajectory, and reflection coefficient [31]. Reference [32] has investigated the outage probability and energy consumption of backscatter-aided UAV networks. Of late, Gang et al. have maximized energy efficiency backscatter-aided UAV networks by optimizing the UAV trajectory, backscatter scheduling, and carrier emitter transmission power [33]. The detailed comparison of different works on backscatter communications is also provided in Table 1.

**Table 1.** Comparison of recent works involving backscatter communications.

| Ref. | Scenario | Objective | UAV/Backscatter | Technique | CSI |
|------|----------|-----------|-----------------|-----------|-----|
| [14] | Single cell | Energy efficiency | Single/single | Dinkelbach and dual theory | Perfect |
| [15] | Single cell | Efficient capacity | None/multiple | Reinforcement learning | Perfect |
| [16] | Single cell | Spectral efficiency | None/single | Dual theory and KKT conditions | Perfect |
| [17] | Single cell | Throughput | None/multiple | Block coordinated descent and successive convex optimization | Perfect |
| [18] | Single cell | Max-min throughput | None/multiple | Block coordinated descent and successive convex optimization | Perfect |
| [19] | Single cell | Secrecy rate | None/single | Dual theory | Perfect |
| [20] | Single | Throughput | None/multiple | Lagrangian method | Perfect |
| [21] | Single cell | Energy efficiency | None/multiple | Alternating optimization and Dual theory | Perfect |
| [22] | Single cell | Spectral efficiency | None/single | Dual theory and KKT conditions | Perfect |
| [23] | Single cell | Outage probability and intercept probability | None/single | Asymptotic expressions and approximate expressions | Perfect |
| [24] | Multi cell | Energy efficiency | None/multiple | Dinkelbach and dual theory | Perfect |
| [25] | Multi cell | Utility | None/multiple | Reinforcement learning and supervised deep learning | Perfect |
| [26] | Multi cell | Interference management | None/multiple | Reinforcement learning | Perfect |
| [27] | Single cell | Throughput | Single/single | Block coordinated descent and successive convex optimization | Perfect |
| [28] | Single cell | Throughput | Single/multiple | Exhaustive search | Perfect |
| [29] | Single cell | Throughput | Single/single | Block coordinated descent and successive convex optimization | Perfect |
| [30] | Single cell | BER and Energy efficiency | multiple/multiple | Closed-form solution for BER | Perfect |
| [31] | Single cell | Secrecy rate | Single/multiple | Block coordinate descent | Perfect |
| [32] | Single cell | Energy efficiency | Single/multiple | Golden section | Perfect |
| [33] | Single cell | Energy efficiency | Single/multiple | Block coordinated decent | Perfect |
| [Our] | Multi cell | Energy efficiency | Multiple/multiple | Sub-gradient and alternating optimization | Imperfect |

### 1.2. Motivation and Contributions

Even though extensive research studies have been done on NOMA ABC-aided networks [14–26], most of these studies have investigated various optimization problems in single-cell terrestrial scenarios. In single-cell terrestrial scenarios, the optimization problems can be convex/concave because there is no inter-cell interference which makes it non-convex. The solution is easy to obtain using convex optimization methods in those cases. However, more practical communication scenarios consist of multi-cell networks, where each cell has a different size and transmit power and can cause co-channel interference to other cells. Considering these challenges, some works have studied multi-cell scenarios with inter-cell interference, where they assume that the network's channel information (CSI) is available, which is challenging to achieve in practice. It is important to note that these studies are based on terrestrial scenarios and do not consider UAV communications. Some recent works have studied backscatter-aided UAV networks [27–33], only one work considers NOMA communication. Moreover, most of the works do not consider ambient backscatter communication (ABC) in their proposed frameworks.

Based on our recent literature study, the work of resource allocation that considers the NOMA ABC-aided UAV network under the assumption of imperfect channel information has not yet been investigated . This paper proposes a new resource allocation framework for the NOMA ABC-aided UAV network to bridge this open gap. Specifically, the power allocation at different UAVs and the reflection power of the backscatter sensor tag (BST) are simultaneously optimized to improve total achievable energy efficiency under imperfect channel information. Due to the inter-UAV interference, the optimization framework is formulated as non-convex, which involves high complexity. Thus, we transform it into convex optimization and then obtain an efficient solution based on a sub-gradient approach. Numerical results are also plotted to check the advantages of the proposed NOMA ABC-aided UAV network against the conventional NOMA UAV network without ABC. The main contributions of this work can be summarized as follow:

1.  A NOMA ABC-aided UAV network is considered in which multiple UAVs transmit superimposed signals to the IoT in their coverage areas using a NOMA protocol. Meanwhile, BSTs also receive UAV signals, add valuable data, and reflect it towards the IoT. On the receiver side,the IoT with strong channel conditions applies SIC to decode the signal received from the UAV and the BST by subtracting the information of other IoT with poor channel conditions. However, the last IoT cannot apply SIC and decode the received signal from the UAV and the BST directly by treating the signal of the first IoT as noise. This work aims to minimize the power consumption of the NOMA ABC-aided UAV network by efficiently allocating the available system resources under the assumption of imperfect channel information.

2.  The formulated power minimization problem is not convex due to inter-UAV interference among UAVs, NOMA interference, and interference of imperfect CSI. Thus, solving the problem directly and obtaining the optimal solution is challenging and complex. Therefore, we transform it and then adopt a sub-gradient approach for an efficient solution. According to this method, all the optimization variables and multipliers are updated in each iteration until convergence. For comparison, we proposed an optimization framework for the traditional NOMA UAV network without involving ABC.

3.  To validate our proposed framework, we obtain numerical results using Monte Carlo simulations for the NOMA ABC-aided UAV network and the benchmark NOMA UAV network without ABC. Results demonstrate the benefits of the ABC-aided NOMA UAV network compared to the benchmark network. Moreover, results also show the effect of imperfect channel information on the overall system's achievable energy efficiency. Furthermore, the impact of other optimization parameters , such as minimum quality of services of individual IoT, transmit power of each UAV, and the number of UAVs, is also depicted. The results reveal that using ABC in NOMA

UAV networks can significantly improve the total achievable energy efficiency of the system.

*1.3. Paper Organization*

The rest of the work is structured as follow. Section 2 provides the details of the proposed system model, i.e., various assumptions and considerations. It also presents the total power minimization problem and discusses different practical constraints. Section 3 presents an efficient solution for the optimization problem based on the sub-gradient method. Section 4 presents and explains the numerical results based on Monte Carlo simulations. Finally, Section 5 concludes our paper with a few exciting future issues and research gaps.

## 2. System Model and Problem Formulation

We consider a NOMA ABC-aided UAV network as shown in Figure 1, where *M* UAVs communicate with IoT using a downlink NOMA protocol. Meanwhile, *N* BSTs in the geographical area of UAVs also receive superimposed signals. More specifically, a BST (denoted as *n*-th BST) utilizing the superimposed signal received from *m*-th UAV, adding useful information on it and then reflecting towards the IoT. This work assumes that: all the devices use an omnidirectional antenna; the CSI is imperfect; UAVs can adjust their heading with a fixed altitude *H*; the horizontal and vertical coordinates of each UAV are represented as $\mathbf{f} \in \{X_m, Y_m\}$.

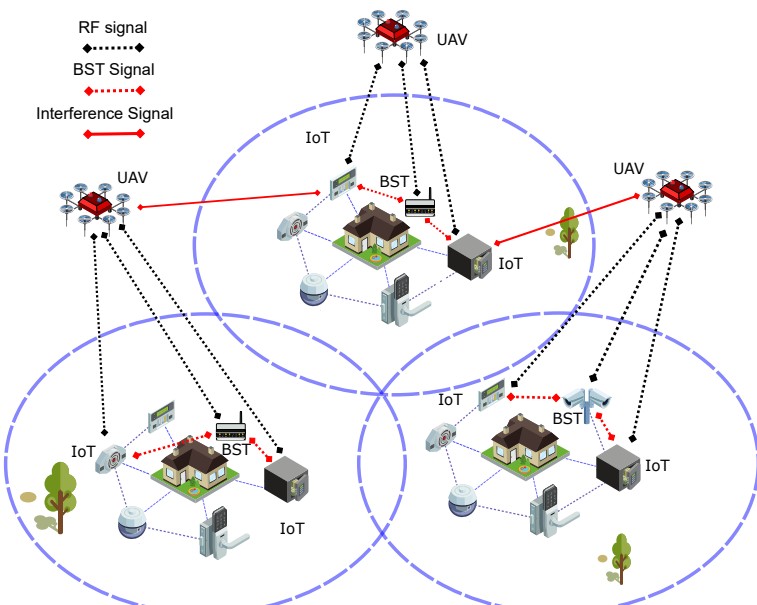

**Figure 1.** System model of NOMA backscatter-aided UAV network

For simplicity, this work assumes that each UAV communicates with two IoT (*i* and *j*) at any given time. Let us denote the transmit power of *m*-th UAV as $P_m$, the power allocation coefficient of *i*-th and *j*-th IoT over *m*-th UAV as $\hat{\varrho}_{i,m}$ and $\hat{\varrho}_{j,m}$, respectively. Then, the superimposed signal of *m*-th UAV, i.e., $x_m$ can be expressed as:

$$x_m = \sqrt{P_m \hat{\varrho}_{i,m}} x_{i,m} + \sqrt{P_m \hat{\varrho}_{j,m}} x_{j,m}, \tag{1}$$

where $x_{i,m}$ and $x_{j,m}$ are the unit power signals of *i*-th IoT and *j*-th IoT. In air to ground UAV communication, the channel is mainly dominated by line of site propagation when the UAV is hovering at moderate altitude [34]. The IoT and BST are considered to be located at

$\mathbf{v}_\iota$ in the horizontal plane under a 2D Cartesian coordinate system, where $\iota \in \{i, j, n\}$. The downlink channel gain from $m$-th UAV to $\iota$-th IoT/BST can be expressed as:

$$g_{\iota,m} = \frac{\alpha_0}{\|\mathbf{f} - \mathbf{v}_\iota\|^2 + H^2} \tag{2}$$

where $\alpha_0$ shows the reference channel gain over a 1 m distance. Next, we model the uplink channel from $n$-th BST to $i$-th and $j$-th IoT associated with $m$-th UAV as $g^u_{n,i,m} = G^u_{i,m} \times d^{\frac{\beta}{2}}_{i,m}$ and $g^u_{n,j,m} = G^u_{j,m} \times d^{\frac{\beta}{2}}_{j,m}$ [35]. Here, $G^u_{i,m}$ and $G^u_{j,m}$ denote the coefficient of Rayleigh fading, $d_{i,m}, d_{j,m}$ represent the distance from $n$-th BST to $i$-th and $j$-th IoT. Moreover, $\beta$ shows the path-loss. Since we consider imperfect CSI, therefore, according to the minimum mean square error model, the above channel models can be estimated as:

$$g_{\iota,m} = \bar{g}_{\iota,m} + \phi_{\iota,m}, \tag{3}$$

$$g^u_{n,i,m} = \bar{g}^u_{n,i,m} + \phi_{n,i,m}, \tag{4}$$

$$g^u_{n,j,m} = \bar{g}^u_{n,j,m} + \phi_{n,j,m}, \tag{5}$$

where $\bar{g}_{\iota,m}$, $\bar{g}^u_{n,i,m}$, and $\bar{g}^u_{n,j,m}$ are the estimated channel gains of $g_{\iota,m}$, $g^u_{n,i,m}$, and $g^u_{n,j,m}$, respectively. Moreover, $\phi_{\iota,m}$, $\phi_{n,i,m}$, and $\phi_{n,j,m}$ denote the estimated channel errors with variance of $\sigma^2_{\phi_{\iota,m}}$, $\sigma^2_{\phi_{n,i,m}}$, and $\sigma^2_{\phi_{n,j,m}}$. For the ease of calculations and discussion, we assume that $\sigma^2_{\phi_{\iota,m}} = \sigma^2_{\phi_{n,i,m}} = \sigma^2_{\phi_{n,j,m}} = \sigma^2_\phi$ in the rest of this paper. The signals that receive at $i$-th IoT and $j$-th IoT from $m$-th UAV can be stated as:

$$\begin{aligned} y_{i,m} &= \bar{g}_{i,m}x_m + \sqrt{\eta_{n,m}}\bar{g}^d_{n,m}\bar{g}^u_{n,i,m}x_m\hat{x}_{n,m} \\ &\quad + \phi_{i,m}x_m + \phi_{n,i,m}\sqrt{\eta_{n,m}} + \omega_{i,m}, \end{aligned} \tag{6}$$

$$\begin{aligned} y_{j,m} &= \bar{g}_{j,m}x_m + \sqrt{\eta_{n,m}}\bar{g}^d_{n,m}\bar{g}^u_{n,j,m}x_m\hat{x}_{n,m} \\ &\quad + \phi_{j,m}x_m + \phi_{n,j,m}\sqrt{\eta_{n,m}} + \omega_{j,m}, \end{aligned} \tag{7}$$

where $\eta_{n,m}$ is the reflection power of $n$-th BST in the coverage area of $m$-th UAV, $\bar{g}^d_{n,m}$ is the channel gain from $m$-th UAV to $n$-th BST, while $\omega_{i,m}$ and $\omega_{j,m}$ are the additive white Gaussian noises. Following the above received signals, the achievable data rate of $i$-th and $j$-th IoT can be stated as:

$$R_{i,m} = \log_2(1 + \gamma_{i,m}) \tag{8}$$

$$R_{j,m} = \log_2(1 + \gamma_{j,m}) \tag{9}$$

where $\gamma_{i,m}$ and $\gamma_{j,m}$ denote the signal to interference plus noise ratios which can be defined as:

$$\gamma_{i,m} = \frac{P_m\hat{\varrho}_{i,m}(\bar{g}_{i,m} + \eta_{n,m}\bar{g}^d_{n,m}\bar{g}^u_{n,i,m})}{\sigma^2_\phi(P_m(\hat{\varrho}_{i,m} + \hat{\varrho}_{j,m}) + \eta_{n,m}) + \xi_{i,m} + \sigma^2} \tag{10}$$

$$\gamma_{j,m} = \frac{P_m\hat{\varrho}_{j,m}(\bar{g}_{j,m} + \eta_{n,m}\bar{g}^d_{n,m}\bar{g}^u_{n,j,m})}{\zeta_{j,m} + \sigma^2_\phi(P_m(\hat{\varrho}_{i,m} + \hat{\varrho}_{j,m}) + \eta_{n,m}) + \xi_{j,m} + \sigma^2} \tag{11}$$

in which $\xi_{\kappa,m} = \bar{g}^{m'}_{\kappa,m}\sum_{m'=1}^M P_{m'}$ is the co-channel interference from other UAVs while $\zeta_{j,m} = P_m\hat{\varrho}_{i,m}(\bar{g}_{j,m} + \eta_{n,m}\bar{g}^d_{n,m}\bar{g}^u_{n,j,m})$ is the NOMA interference after the SIC decoding process.

This work seeks to minimize the total power consumption of UAVs to design an energy efficient transmission framework for the NOMA ABC-aided UAV network. More specifically,

the power of each UAV and reflection of BSts are simultaneously optimized while ensuring the minimum data rate of the individual IoT. Mathematically, it can be formulated as:

$$\mathcal{P}_1: \quad \min_{(\hat{\varrho}_{i,m}, \hat{\varrho}_{j,m}, \eta_{n,m})} \sum_{m=1}^{M} P_m(\hat{\varrho}_{i,m} + \hat{\varrho}_{j,m}) \tag{12}$$

$$s.t. \begin{cases} \mathcal{C}_1: \sum_{m=1}^{M} R_{i,m} \geq R_{i,m}^{min}, \ \forall i, \\ \mathcal{C}_2: \sum_{m=1}^{M} R_{j,m} \geq R_{j,m}^{min}, \ \forall j, \\ \mathcal{C}_3: \sum_{m=1}^{M} p_m(\hat{\varrho}_{i,m} + \hat{\varrho}_{j,m}) \geq P_{max}, \ \forall m, \\ \mathcal{C}_4: \sum_{m=1}^{M} (\hat{\varrho}_{i,m} + \hat{\varrho}_{j,m}) \leq 1, \ \forall i, j, \\ \mathcal{C}_5: \sum_{m=1}^{M} \eta_{n,m} \leq 1, \ \forall n, \end{cases}$$

where $\mathcal{C}_1$ and $\mathcal{C}_2$ guarantee the minimum data rate of $i$-th IoT and $j$-th IoT, where $R_{i,m}^{min}$ and $R_{j,m}^{min}$ denote the minimum thresholds. $\mathcal{C}_3$ limits the total power budget of UAVs, where $P_{max}$ is the maximum power budget. Constraint $\mathcal{C}_4$ ensures the power allocation according to NOMA protocol while $\mathcal{C}_5$ is used for reflection power control.

### 3. Proposed Optimization Solution

This section presents an efficient solution for our optimization problem, formulated in (12). We can see that $\mathcal{P}_1$ is non-convex due to rate constraints which contain interference due to NOMA users, co-channel, and imperfect CSI. Moreover, the formulated problem is coupled with two optimization variables, i.e., transmit power and reflection coefficient, which pose high complexity and make it very challenging to obtain the optimal global solution. Thus, a low complex and energy efficient technique is required and essential. Therefore, an alternating optimization approach can be efficiently adopted here, where the original problem will be decoupled into subproblems for individual optimization variables. Following the work in [36], constraints $\mathcal{C}_1$ and $\mathcal{C}_2$ can be smartly simplified as:

$$\bar{\mathcal{C}}_1 = P_m \hat{\varrho}_{i,m}(\bar{g}_{i,m} + \eta_{n,m} \bar{g}_{n,m}^d \bar{g}_{i,m}^u) \geq (R_{i,m}^{min} - 1)$$
$$(\sigma_\phi^2(P_m(\hat{\varrho}_{i,m} + \hat{\varrho}_{j,m}) + \eta_{n,m}) + \xi_{i,m} + \sigma^2) \tag{13}$$

$$\bar{\mathcal{C}}_2 = P_m \hat{\varrho}_{j,m}(\bar{g}_{j,m} + \eta_{n,m} \bar{g}_{n,m}^d \bar{g}_{j,m}^u) \geq (R_{j,m}^{min} - 1)$$
$$(\zeta_{j,m} + \sigma_\phi^2(P_m(\hat{\varrho}_{i,m} + \hat{\varrho}_{j,m}) + \eta_{n,m}) + \xi_{j,m} + \sigma^2) \tag{14}$$

Next, for any given reflection at BSTs, $\mathcal{P}_1$ can be expressed as:

$$\mathcal{P}_2: \quad \min_{(\hat{\varrho}_{i,m}, \hat{\varrho}_{j,m})} \sum_{m=1}^{M} P_m(\hat{\varrho}_{i,m} + \hat{\varrho}_{j,m})$$
$$s.t. \left\{ \bar{\mathcal{C}}_1, \bar{\mathcal{C}}_2, \mathcal{C}_3, \mathcal{C}_4, \right. \tag{15}$$

where $\mathcal{P}_2$ is now a power allocation problem at UAVs and convex optimization. To efficiently solve this problem, we adopt the sub-gradient method as

$$
\begin{aligned}
\mathcal{L}_2(\lambda_{i,m}, \lambda_{j,m}, \mu_m, \pi_m) = {} & P_m(\hat{\varrho}_{i,m} + \hat{\varrho}_{j,m}) + \lambda_{i,m} \\
& ((R_{i,m}^{min} - 1)(\sigma_\phi^2(P_m(\hat{\varrho}_{i,m} + \hat{\varrho}_{j,m}) + \eta_{n,m}) + \xi_{i,m} + \sigma^2) \\
& - P_m\hat{\varrho}_{i,m}(\bar{g}_{i,m} + \eta_{n,m}\bar{g}_{n,m}^d\bar{g}_{i,m}^u)) + \lambda_{j,m}((R_{j,m}^{min} - 1) \\
& (\zeta_{j,m} + \sigma_\phi^2(P_m(\hat{\varrho}_{i,m} + \hat{\varrho}_{j,m}) + \eta_{n,m}) + \xi_{j,m} + \sigma^2) \\
& - P_m\hat{\varrho}_{j,m}(\bar{g}_{j,m} + \eta_{n,m}\bar{g}_{n,m}^d\bar{g}_{j,m}^u)) + \mu_m(P_{max} \\
& - P_m(\hat{\varrho}_{i,m} + \hat{\varrho}_{j,m})) + \pi_m(1 - (\hat{\varrho}_{i,m} + \hat{\varrho}_{j,m}))
\end{aligned} \tag{16}
$$

where $\mathcal{L}_2(.)$ is the Lagrange function of $\mathcal{P}_2$, and $\lambda_{i,m}, \lambda_{j,m}, \mu_m, \pi_m$ are its associated variables. Now the partial derivation of (16) with respect to power allocation coefficients can be obtained as:

$$
\begin{aligned}
\frac{\partial \mathcal{L}_2(\lambda_{i,m}, \lambda_{j,m}, \mu_m, \pi_m)}{\partial \hat{\varrho}_{i,m}} = {} & P_m(1 - \bar{g}_{i,m}\lambda_{i,m} - \eta_{n,m}\bar{g}_{i,m}^u \\
& \bar{g}_{n,m}^d\lambda_{i,m} + (R_{i,m}^{min} - 1)(\lambda_{i,m} + \lambda_{j,m})\sigma_\phi^2 + \mu_m),
\end{aligned} \tag{17}
$$

$$
\begin{aligned}
\frac{\partial \mathcal{L}_2(\lambda_{i,m}, \lambda_{j,m}, \mu_m, \pi_m)}{\partial \hat{\varrho}_{j,m}} = {} & P_m(1 - \bar{g}_{j,m}\lambda_{j,m} - \eta_{n,m}\bar{g}_{j,m}^u \\
& \bar{g}_{n,m}^d\lambda_{i,m} + (R_{i,m}^{min} - 1)(\lambda_{i,m} + \lambda_{j,m})\sigma_\phi^2 + \mu_m).
\end{aligned} \tag{18}
$$

Accordingly, for the given values of $\hat{\varrho}_{i,m}$ and $\hat{\varrho}_{j,m}$, the problem $\mathcal{P}_1$ can be simplified as

$$
\mathcal{P}_3: \quad \min_{(\eta_{n,m})} \sum_{m=1}^{M} P_m(\hat{\varrho}_{i,m} + \hat{\varrho}_{j,m}) \\
s.t. \left\{ \bar{\mathcal{C}}_1, \bar{\mathcal{C}}_2, \mathcal{C}_5, \right. \tag{19}
$$

where $\mathcal{P}_3$ is the problem of efficient reflection power design at BSTs which can be solved through the sub-gradient method as:

$$
\begin{aligned}
\mathcal{L}_3(\lambda_{i,m}, \lambda_{j,m}, \alpha_{n,m}) = {} & P_m(\hat{\varrho}_{i,m} + \hat{\varrho}_{j,m}) + \lambda_{i,m} \\
& ((R_{i,m}^{min} - 1)(\sigma_\phi^2(P_m(\hat{\varrho}_{i,m} + \hat{\varrho}_{j,m}) + \eta_{n,m}) + \xi_{i,m} + \sigma^2) \\
& - P_m\hat{\varrho}_{i,m}(\bar{g}_{i,m} + \eta_{n,m}\bar{g}_{n,m}^d\bar{g}_{i,m}^u)) + \lambda_{j,m}((R_{j,m}^{min} - 1) \\
& (\zeta_{j,m} + \sigma_\phi^2(P_m(\hat{\varrho}_{i,m} + \hat{\varrho}_{j,m}) + \eta_{n,m}) + \xi_{j,m} + \sigma^2) - \\
& P_m\hat{\varrho}_{j,m}(\bar{g}_{j,m} + \eta_{n,m}\bar{g}_{n,m}^d\bar{g}_{j,m}^u)) + \alpha_{n,m}((\eta_{n,m} - 1)
\end{aligned} \tag{20}
$$

where $\mathcal{L}_3(.)$ is the Lagrange function of $\mathcal{P}_3$, and $\varrho_{n,m}$ is the Lagrange multiplier. Now we compute the partial derivation as:

$$
\begin{aligned}
\frac{\partial \mathcal{L}_3(\lambda_{i,m}, \lambda_{j,m}, \alpha_{n,m})}{\partial \eta_{n,m}} = {} & \hat{\varrho}_{i,m}\bar{g}_{i,m}\lambda_{i,m}P_m - \hat{\varrho}_{j,m}\bar{g}_{j,m}\lambda_{i,m} \\
& \lambda_{j,m}P_m + (R_{i,m}^{min} - 1)(\lambda_{i,m} + \lambda_{j,m})\sigma_\phi^2 + \alpha_{n,m}).
\end{aligned} \tag{21}
$$

Finally, the optimization variables and Lagrange multipliers are iteratively updated as:

$$
\hat{\varrho}_{i,m}(\theta + 1) = \left[ \hat{\varrho}_{i,m}(\theta) - \varphi(\theta)\frac{\partial \mathcal{L}_2(.)}{\partial \hat{\varrho}_{i,m}} \right]^+ \tag{22}
$$

$$\hat{\varrho}_{j,m}(\theta+1) = \left[\hat{\varrho}_{j,m}(\theta) - \varphi(\theta)\frac{\partial\mathcal{L}_2(.)}{\partial\hat{\varrho}_{j,m}}\right]^+ \tag{23}$$

$$\eta_{n,m}(\theta+1) = \left[\eta_{n,m}(\theta) - \varphi(\theta)\frac{\partial\mathcal{L}_3(.)}{\partial\eta_{n,m}}\right]^+ \tag{24}$$

$$\lambda_{i,m}(\theta+1) = \left[\lambda_{i,m}(\theta) - \varphi(\theta)\Big(P_m\hat{\varrho}_{i,m}(\bar{g}_{i,m} + \eta_{n,m}\right.$$
$$\bar{g}_{n,m}^d\bar{g}_{i,m}^u) - (R_{i,m}^{min} - 1)(\sigma_\phi^2(P_m$$
$$\left.(\hat{\varrho}_{i,m} + \hat{\varrho}_{j,m}) + \eta_{n,m}) + \xi_{i,m} + \sigma^2)\Big)\right]^+, \tag{25}$$

$$\lambda_{j,m}(\theta+1) = \left[\lambda_{j,m}(\theta) - \varphi(\theta)\Big(P_m\hat{\varrho}_{j,m}(\bar{g}_{j,m} + \eta_{n,m}\right.$$
$$\bar{g}_{n,m}^d\bar{g}_{j,m}^u) - (R_{j,m}^{min} - 1)(\zeta_{j,m} + \sigma_\phi^2(P_m$$
$$\left.(\hat{\varrho}_{i,m} + \hat{\varrho}_{j,m}) + \eta_{n,m}) + \xi_{j,m} + \sigma^2)\Big)\right]^+, \tag{26}$$

$$\mu_m(\theta+1) = \left[\mu_m(\theta) - \varphi(\theta)\Big(P_{max} - p_m(\hat{\varrho}_{i,m} + \hat{\varrho}_{j,m})\Big)\right]^+ \tag{27}$$

$$\pi_m(\theta+1) = \left[\pi_m(\theta) - \varphi(\theta)\Big(1 - (\hat{\varrho}_{i,m} + \hat{\varrho}_{j,m})\Big)\right]^+ \tag{28}$$

$$\alpha_{n,m}(\theta+1) = \left[\alpha_{n,m}(\theta) - \varphi(\theta)\Big(1 - \eta_{n,m}\Big)\right]^+ \tag{29}$$

where $\theta$ indicates iteration number, and $\varphi \geq 0$ is the step size, respectively. The proposed iterative sub-gradient method can be updated in each iteration until convergence of the transmit power and reflection coefficient.

## 4. Numerical Results and Discussion

In this section, we provide and discuss numerical results obtained from the solution provided in the previous section. This work compares the proposed NOMA ABC-aided UAV network (denoted as ABC-enhanced NOMA) with the traditional pure NOMA UAV network (stated as Pure NOMA), where ABC is not considered. For simulation, the maximum number of UAVs is considered as 10. The imperfect CSI parameter is 0.01 and 0.001. The individual data rate of IoT varies from 0.2 to 0.9 b/s. Moreover, the maximum transmit power of each UAV is 30 dBm. Each UAV altitude is fixed at $H = 10$ m, and the path loss exponent is set as 5 dBm. The achievable energy efficiency of the system is calculated as the ratio of the sum data rate to the total power consumption (including circuit power).

Figures 2 and 3 show the achievable energy efficiency versus individual data rate of IoT for different numbers of UAVs and imperfect CSI parameters. In particular, the number of UAVs in Figure 2 is considered as 5 and 10, while the values of the imperfect parameter in Figure 3 are varied from 0.001 to 0.01, respectively. In both figures, the minimum data rate per IoT varies from 0.2 to 0.9 b/s, and the maximum power budget of a UAV is set as 30 dBm. Moreover, the value of the imperfect CSI parameter in Figure 2 is set to 0.001 while the number of UAVs in Figure 3 is set as 10. It can be evident from both figures that the proposed NOMA ABC-aided UAV network achieves higher energy efficiency than the traditional NOMA UAV network without ABC. This is because ABC adds useful gain to the received signal of IoT without consuming any additional power at UAVs, which shows the benefit of integrating the ABC system into NOMA UAV networks. We can also observe that the system with more UAVs in Figure 2 obtains higher achievable energy efficiency than the system with fewer UAVs. A system with more UAVs can have more BSTs and accommodate more IoT, resulting in extra energy efficiency. Similarly, a system with a lower value of imperfect CSI, as shown in Figure 3 can achieve higher performance than

the system with high values. Furthermore, it can also be noted that the achievable energy efficiency of the system reduces when the minimum quality of services of individual IoT increases. This is because IoT with high data rate demands requires more transmit power at UAVs, which affects the overall system's energy efficiency. In both figures, the proposed NOMA ABC-aided UAV network significantly outperforms its counterpart benchmark pure NOMA UAV network.

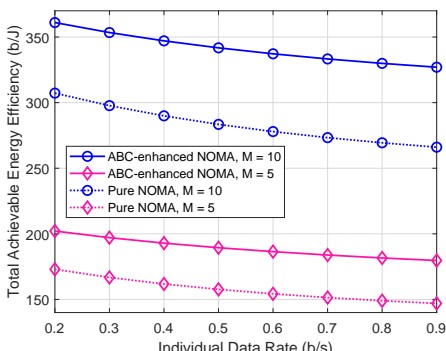

**Figure 2.** The minimum data rate of IoT versus achievable energy efficiency of the proposed NOMA ABC-aided UAV network and the benchmark traditional NOMA UAV network without ABC. Here we plot results with different numbers of UAVs.

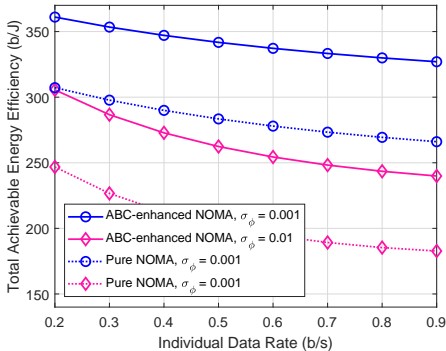

**Figure 3.** The minimum data rate versus achievable energy efficiency of the NOMA ABC-aided UAV network and the traditional pure NOMA UAV network without ABC. Here we plot results with different values of $\sigma_\phi$.

Next, we study the achievable energy efficiency of the system against the varying transmit power of the UAV for different numbers of UAVs in the system, as shown in Figure 4, and for different values of imperfect CSI parameter, as shown in Figure 5. In both figures, the achievable energy efficiency first increases with the increase in the transmit power of UAVs and then remains unchanged with a further increase in the available power. For instance, when the UAV in each coverage area is transmitting with 10 dBm, the achievable energy efficiency of the NOMA UAV network is 330 b/j, and it reaches 370 b/j as the power of the UAVs approaches 15 dBm. The system performance remains unchanged when the allocated power UAVs exceeds 15 dBm. However, in both figures, the proposed NOMA ABC-aided UAV network achieves significantly high energy efficiency than the traditional pure NOMA UAV network. It shows the importance of ABC system for low-powered IoT in future UAV networks. Moreover, the system with more UAVs in Figure 4 achieves higher achievable energy efficiency than the system with fewer UAVs. Accordingly, the system with lower values of imperfect CSI in Figure 6 obtains more energy efficiency compared to the system having higher values of imperfect CSI.

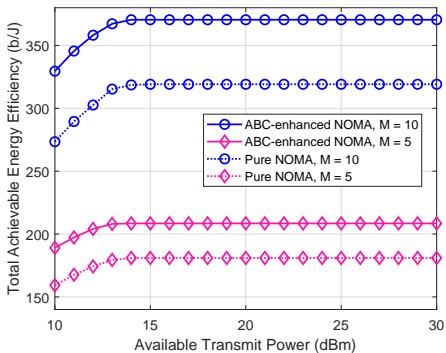

**Figure 4.** The available transmit power at UAVs versus achievable energy efficiency of the NOMA ABC-aided UAV network. Here we plot results with different number of UAVs.

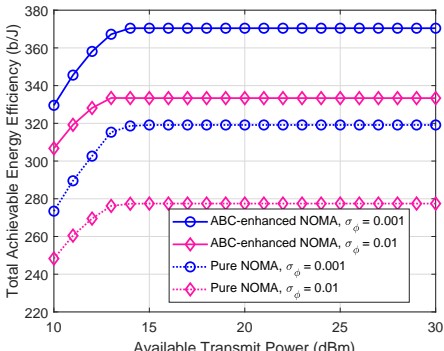

**Figure 5.** The available transmit power at UAVs versus achievable energy efficiency of the NOMA ABC-aided UAV network. Here we plot results with different values of $\sigma_\phi$.

Finally, Figure 6 discusses the system's achievable energy efficiency against the different values of IoT minimum data rate when changing the values of circuit power (i.e., $P_c$) consumption. In this figure, we set two circuit power consumption such as $P_c = 5$ dBm and $P_c = 10$ dBm, respectively. We can see that the system with higher $P_c$ consumption provides lower achievable energy efficiency than the one with lower $P_c$ consumption. However, we can see that the proposed NOMA ABC-aided UAV network performs significantly better in achievable energy efficiency than the pure NOMA UAV network. Moreover, system performance reduces as the quality of services of IoT increases. This is because high data rate demands require more transmit power at UAVs, which affects the system's achievable energy efficiency.

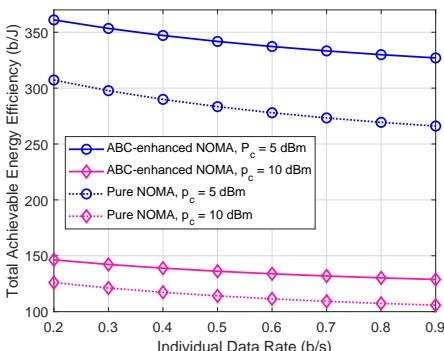

**Figure 6.** The minimum data rate of IoT versus achievable energy efficiency of the NOMA ABC-aided UAV network and the traditional pure NOMA UAV network without ABC system. Here we plot results with different values of $P_c$.

## 5. Conclusions

NOMA and ABC are promising technologies for next-generation UAV networks to connect low-powered IoT. This paper has provided a new optimization framework for a NOMA ABC-aided UAV network to minimize the total power consumption under imperfect CSI. The optimization problem has been formulated as non-convex, hence challenging to obtain the optimal global solution. Therefore, it has been solved through an alternating optimization approach, where the original problem is coupled into subproblems. Then, a sub-gradient method is adopted to obtain a low complexity, efficient solution. A benchmark pure NOMA UAV optimization framework without ABC has also been investigated for comparison. Simulation results have proved the significance of the proposed NOMA ABC-enabled UAV NOMA network compared to the benchmark pure NOMA UAV network without the ABC system. Our work can be enhanced in several directions. For example, intelligent reflecting surfaces can be combined with ABC-enabled NOMA UAV networks. Another potential topic is to investigate the physical layer security of such networks.

**Author Contributions:** Conceptualization, W.U.K. and A.M.; methodology, W.U.K. and S.A. and A.M.; formal analysis, F.N.A.-W. and H.A. and M.A.D. and M.K.N.; investigation, W.U.K. and A.M.; supervision, K.R. and T.S. and S.A.; project administration, S.A. and K.R. and T.S.; writing—original draft preparation, S.A. and W.U.K; writing—review and editing, S.A. and W.U.K. and F.N.A.-W.-W. and H.A.; investigation, validation, writing—review and editing, W.U.K. and S.A. and M.A.D. and M.K.N.; data curation, A.M.; visualization and funding acquisition, K.R. and T.S. All authors have read and agreed to the published version of the manuscript.

**Funding:** The authors extend their appreciation to the Deanship of Scientific Research at King Khalid University for funding this work through Large Groups Project under grant number (18/43). Princess Nourah bint Abdulrahman University Researchers Supporting Project number (PNURSP2022R303), Princess Nourah bint Abdulrahman University, Riyadh, Saudi Arabia. The work would like to thank the Deanship of Scienctific Research at Umm Al-Qura University for supporting this work by grant code: (22UQU4310373DSR25).

**Institutional Review Board Statement:** Not applicable.

**Informed Consent Statement:** Not applicable.

**Data Availability Statement:** Not applicable.

**Conflicts of Interest:** The authors declare no conflict of interest.

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
