# Peer review of "Energy Efficient Transmission Design for NOMA Backscatter-Aided UAV Networks with Imperfect CSI"

_drones, doi:10.3390/drones6080190_

Round 1

Reviewer 1 Report

The English writing, organization, and presentation quality of this work are really nice. Abstract and Conclusion provide a very clear idea about this work and the authors’ contributions. This paper has proposed backscatter-enabled UAV communication using the NOMA technique. In particular, the authors have maximized the achievable energy efficiency of the system through optimal power allocation and reflection coefficient. This work looks very interesting and timely. The presented results validate the significance of this work. I have noticed a few issues that authors should consider:

1.     In the Introduction, it is highly recommended to provide a Table for recent works on Backscatter Communication as readers can fastly understand the contributions from the Table.

2.  What is the difference between backscatter and intelligent reflecting surfaces? Any comparative Table?

3.     What do you think backscatter communication is more efficient than traditional relay systems?

4.     Figure 2: It is suggested to make the description precise and add relevant information in the above relevant section.

5.     Line 27-28: What is RF, WiFi? Authors should carefully check each acronym and define in the first place of appearance for better readability.

6.     Some variables are not defined. So, equations are needed to be further explained for the readers of this work. The authors should recheck all equations and define each variable as it can cause difficulty for anyone who is not much familiar with this domain.

7.     Conclusion section is precise and well-written.  

8.     I am looking forward to PLS of the proposed network.

9.     How this network can be further extended? For instance, how to increase the number of users in the coverage area of UAV? 

10.     Although references are updated, but more recent works need to be studied and reported.

Author Response

Response to Reviewer 1 Comments

Point 1: The English writing, organization, and presentation quality of this work are really nice. Abstract and Conclusion provide a very clear idea about this work and the authors’ contributions. This paper has proposed backscatter-enabled UAV communication using the NOMA technique. In particular, the authors have maximized the achievable energy efficiency of the system through optimal power allocation and reflection coefficient. This work looks very interesting and timely. The presented results validate the significance of this work. I have noticed a few issues that authors should consider:

Response 1: We thank this reviewer for thorough review of our manuscript. We have improved the quality of literature review and have made our contributions more clear. We have also added the table of comparison on different backscatter communication works in the introduction section. We report our response and the revisions in the following text.

Point 2: In the Introduction, it is highly recommended to provide a Table for recent works on Backscatter Communication as readers can fastly understand the contributions from the Table.

Response 2: Thank you for suggesting the improvement. We have added a Table 1 of comparison on different backscatter communication works to the revised manuscript. Please refer to the revised paper.

Point 3: What is the difference between backscatter and intelligent reflecting surfaces? Any comparative Table?

Response 3: Thank you so much for the comments. In this work, we consider backscatter communication which completely different technology from intelligent reflecting surfaces. Therefor, it is not appropriate to add the comparative table here.

Point 4: What do you think backscatter communication is more efficient than traditional relay systems?

Response 4: We thank the reviewer for the helpful comment. A traditional relay requires battery power for transmission and less energy efficient. However, backscatter communication do not require a dedicated battery and uses the existing RF signals for circuit operation and data transmission.

Point 5: Figure 2: It is suggested to make the description precise and add relevant information in the above relevant section.

Response 5: Thank you very much for the comments. We have checked all the descriptions of different figure and added information to those where needed.

Point 6: Line 27-28: What is RF, WiFi? Authors should carefully check each acronym and define in the first place of appearance for better readability.

Response 6: Thank you for the useful comment, we have now updated this sentence. Moreover, we have define all abbreviations where needed. Please refer to the revised paper.

Point 7: Some variables are not defined. So, equations are needed to be further explained for the readers of this work. The authors should recheck all equations and define each variable as it can cause difficulty for anyone who is not much familiar with this domain.

Response 7: Thank you so much for carefully reviewing our paper. We have carefully checked all the equations and formulas and define every variable properly. Please refer to the revised paper.

Point 8: Conclusion section is precise and well-written.

Response 8: Thank you so much for the positive comments.

Point 9: I am looking forward to PLS of the proposed network.

Response 9: Thank you for your suggestion and identifying a very potential problem. We plan to investigate the PLS of the proposed model in future studies.

Point 10: How this network can be further extended? For instance, how to increase the number of users in the coverage area of UAV?

Response 9: Thank you so much for the comments. Our work can be easily extended to large-scale network. For instance, if we consider multi-carrier communication and reuse it by all UAVs. In this way, each carrier will accommodate two NOMA users. Therefore if we have $N$ number of carriers, each UAV can serves $N\times 2$ users. 

Point 11: Although references are updated, but more recent works need to be studied and reported.

Response 9: Thank you so much for the comments. In this regards, we would like to clarify that we have already added all the recent works on backscatter aided NOMA UAV communication. Yet to address this comment, we have added some more works on NOMA and UAV communication in the revised paper, . Please refer to the revised paper.

Reviewer 2 Report

This paper provides an energy efficient transmission design ABC-aided UAV networks using NOMA. The authors aim to optimize the power consumption of UAV system while ensuring the minimum data rate of IoT. Specifically, the transmit power of UAVs and the reflection coefficient of ABC system are simultaneously optimized under the assumption of imperfect channel state information (CSI). Due to co-channel interference among UAVs, imperfect CSI, and NOMA interference, the joint optimization problem is formulated as non-convex, which involves high complexity and makes it hard to obtain the optimal solution. Thus, it is first transformed and then solved by a sub-gradient method with low complexity. In addition, a conventional NOMA UAV framework is also studied for comparison without involving ABC. Numerical results demonstrate the benefits of using ABC in a NOMA UAV network compared to the conventional UAV framework.

This work is very well written, and the idea of NOMA, UAV and backscatter communication is very timely and good. Moreover, the contributions are solid. There are some comments that can further improve the quality of this paper:

1) The paper should be checked for all typos and grammar errors and need to correct those where needed.

2) All the mathematical equations and formulas need to be checked and all the parameters should be defined properly on first place.

3) How can backscatter communication improve the energy efficiency of such network?

4) What is the effect of UAV height on the performance of backscatter communications?

5) How this work can be extended further?

6) Some recent work on NOMA and backscatter communication can be added.

Author Response

Response to Reviewer 2 Comments

Point 1: This work is very well written, and the idea of NOMA, UAV and backscatter communication is very timely and good. Moreover, the contributions are solid. There are some comments that can further improve the quality of this paper:

Response 1: We thank this reviewer for  thorough review of our manuscript. We have discussed the mentioned works in the updated manuscript.

Point 2: The paper should be checked for all typos and grammar errors and need to correct those where needed.

Response 2: Thank you for suggesting the improvement. An extensive proofreading has been done for typos and grammar errors and corrected those where needed.

Point 3: All the mathematical equations and formulas need to be checked and all the parameters should be defined properly on first place.

Response 3: Thank you so much for the comments. We have double checked all the equations and formulas and properly explain everyone.

Point 4: How can backscatter communication improve the energy efficiency of such network?

Response 4: We thank the reviewer for the helpful comment. Backscatter communication provides very high energy and spectral efficiency because it uses the existing radio frequency (RF) signals, i.e., WiFi, TV, cellular base station instead of dedicated battery power. Therefore, backscatter communication is the concept of providing battery-free transmission among different wireless devices in future wireless networks. Based on backscatter principle, with the help of sensors, it harvests energy using the existed signals for circuit operation. Then add its own data on the signal and reflect it towards nearby users.

Point 5: What is the effect of UAV height on the performance of backscatter communications?

Response 5: Thank you very much for the comments. In this regard, we would like to clarify that the UAV height do not have any effect on the performance of backscatter communication as long as the UAV provide line of site communication. If the link between backscatter device and UAV is non line of site, it might effect the performance of the backscatter communication. For instance, if backscatter is receiving very weak signal from UAV, the harvested energy might not efficient for the backscatter communication and it can be effected by energy outage.

Point 6: How this work can be extended further?

Response 6: Thank you for the useful comment, our work can be extended in several ways. To further enhance the performance of the system, the trajectory different UAVs can be optimized. Besides that, if the users are randomly distributed in the coverage area of UAVs, user association to different UAVs can be optimized to further enhance the system energy efficiency. Further, our system can also be extended to multi-carrier communications.

Point 7: Some recent work on NOMA and backscatter communication can be added.

Response 7: Thank you for the useful comment. We have added some works on NOMA and UAV communications in the revised manuscript.